# Exploring Photonic Crystals: Band Structure and Topological Interface States

**Melquiades de Dios-Leyva [1], Andy Márquez-González [1] and Carlos Alberto Duque [2,*]**

1   Department of Theoretical Physics, University of Havana, San Lázaro y L, Vedado, Havana 10400, Cuba;
    dedios@fisica.uh.cu (M.d.D.-L.); marquezandy803@gmail.com (A.M.-G.)
2   Grupo de Materia Condensada-UdeA, Instituto de Física, Facultad de Ciencias Exactas y Naturales,
    Universidad de Antioquia UdeA, Calle 70 No. 52-21, Medellín 50010, Colombia
*   Correspondence: carlos.duque1@udea.edu.co

**Abstract:** The physical mechanisms supporting the existence of topological interface modes in photonic structures, formed with the concatenation of two finite, $N$-period, one-dimensional photonic crystals, are investigated. It is shown that these mechanisms originate from a specific configuration of bands and bandgaps of topological origin in the band structure of the concatenated structure. Our analysis reveals that the characteristics of such a configuration depend on the structural parameters, including the number, $N$, of unit cells, and determine the properties of the corresponding resonant transmission peak. It was shown that the width and maximum value of the transmission peaks decrease with $N$. These results not only provide new physical insight into the origin and nature of such modes, but also can be used to control and manipulate the transmission peak properties, such as peak values, full width at half maximum (FWHM), and $Q$-factor, which are of special interest in the fields of optical sensing, filters, etc.

**Keywords:** finite crystals; electromagnetic wave propagation; topological interface states; photonic structures

## 1. Introduction

Since the seminal work of Haldane and Raghu [1], many theoretical and experimental works have been devoted to the study of topological phenomena in photonic crystals [2], which are artificial periodic arrays of materials with different refractive indices in one, two, and three dimensions. This interest is motivated by the fact that photonic crystals offer the possibility of controlling and manipulating the properties of light, a possibility closely related to the existence of photonic bandgaps in the dispersion relation of these structures. Thus, if the frequency of a light beam is in the range of a bandgap, the beam cannot propagate inside the structure, leading to remarkable features that have applications in many photonic devices, such as optical isolators, topological lasers, tunable filters, and resonators [3–6].

Recently, special attention has been paid to the study of topological interface states that can be formed at the boundary between two one-dimensional (1D) photonic crystals (PCs) having overlapping bandgaps [7–16]. Although the existence of interface states on such a boundary was first predicted by Kavokin et al. [7], the connection between these modes, called optical Tamm states, and topological concepts was shown by Xiao et al. [8]. Now, there are two aspects of fundamental importance in these studies. One of them refers to the existence of such states and their topological nature. In contrast, the second one is related to the importance of their properties from a fundamental and practical point of view. The first aspect is implemented by using the electromagnetic and topological properties of the constitutive PCs. While the existence is guaranteed when the surface impedance on both sides of the structure is of an opposite sign [8], the topological properties are characterized by the Zak phases [15] associated with the band structure in each PC, which

is directly related to the surface impedance [8]. For PCs with inversion symmetry, which will be the focus of our attention, the Zak phase is a topological invariant that takes the quantized values $0$ or $\pi$ if the origin of coordinates is chosen at the inversion center.

Furthermore, the properties of the topological interface states can be obtained by studying the optical transmission spectrum's corresponding resonant structure and the electric field's distribution around the interface between the two PCs. The former refers to the transmission through the structure formed by concatenating the constituting PCs (combined structure). These procedures have not only been used to verify the existence of the considered states but also for studying the quantitative characteristics of quantities that are of particular importance in applications such as transmission peak values, full width at half maximum (FWHM), the *Q*-factor, etc. [13,16]. Although these studies have given useful information about the properties of the topological interface modes, the physical mechanisms supporting their existence and determining their main characteristics have not been reported. It is the purpose of this work to explore the origin and nature of these mechanisms.

In contrast to the existent approach, in which the condition for the existence of an interface state has been established by using the topological properties of the constituting PCs, here we focus on the role played by the band structure of the concatenated PCs. It is shown, then, that the physical mechanism supporting the existence of the interface states originates from a specific configuration of bands and bandgaps of topological origin in the mentioned band structure. This result provides a novel understanding of the topological interface states and an efficient procedure for controlling their properties, as shown below.

## 2. Theory

To carry out the study, we will focus our attention on the propagation of *TE* modes in the photonic structure *S* shown schematically in Figure 1a, which is constructed with the concatenation of two finite, *N*-period, photonic crystals [17–19], namely, *PC*1 and *PC*2, where the former and latter materials occupy the left and right parts of the structure, respectively, i.e., *S* = *PC*1 | *PC*2. In the following, the calculations will be performed by using the transfer-matrix method, which can be easily implemented numerically.

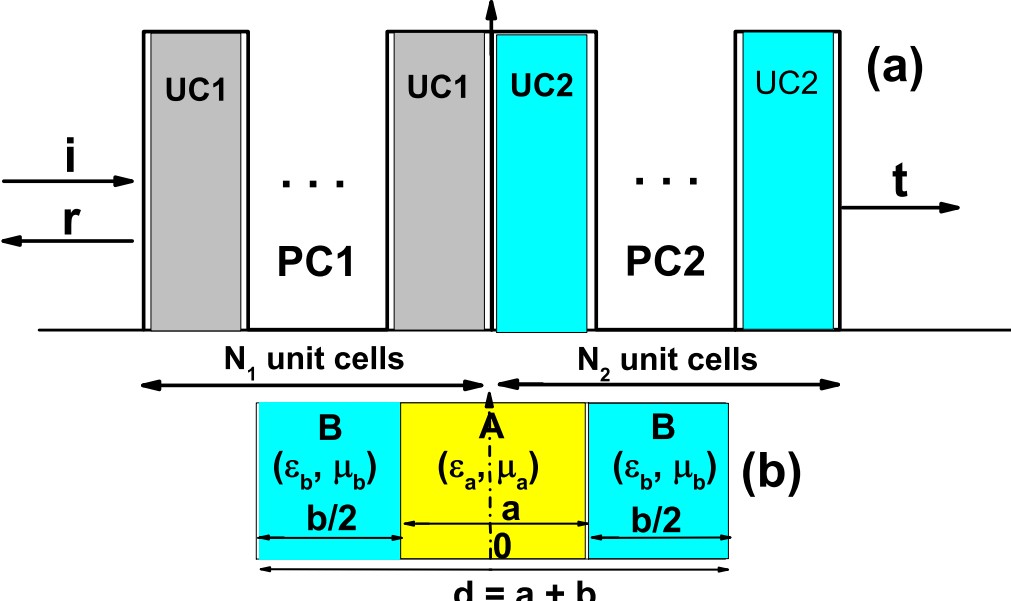

**Figure 1.** (**a**) Photonic structure formed with the concatenation of two finite, *N*-period, photonic crystals *PC*1 and *PC*2 composed of $N_1$ and $N_2$ unit cells, respectively. Horizontal arrows indicate the incident (*i*), reflected (*r*), and transmitted (*t*) coefficients. (**b**) Each unit cell is composed of layers *A* and *B* with permittivity and permeability $(\varepsilon_a, \mu_a)$ and $(\varepsilon_b, \mu_b)$. The period is given by $d = a + b$, where *a* and *b* are the widths of layers *A* and *B*, respectively.

To obtain the energy spectrum of the finite combined structure $S$, we adopt the procedure in which $S$ is taken as the unit cell of a 1D infinite periodic superlattice. As shown in different calculations [20,21], this procedure appropriately describes the energy spectrum of any finite structure, especially when its size is large enough. Thus, if $M$ is the corresponding transfer matrix through this unit cell and $\omega$ is the frequency of the electromagnetic field, the expression determining the band structure is given by

$$\cos \beta = \frac{1}{2} Tr(M) = F(\omega),\tag{1}$$

where $\beta = KD$ is the Bloch phase, $K$ the Bloch wavevector, $D$ the period or unit cell size, and the frequency-dependent function $F(\omega)$ also depends on the physical and geometrical parameters of the structure.

The next step is to express the matrix $M$ in terms of the transfer matrices $M_1$ and $M_2$ through $PC1$ and $PC2$, respectively, which are given by [17,18]

$$M_j = W_j^{N_j} = \frac{\sin N_j \beta_j}{\sin \beta_j} W_j - \frac{\sin(N_j - 1)\beta_j}{\sin \beta_j} I,\tag{2}$$

with $j = 1, 2$, where $W_j$ is the transfer matrix through the unit cell of $PCj$ (see Figure 1b), $I$ the $2 \times 2$ unit matrix, $N_j$ the number of unit cells of $PCj$, and $\beta_j$ the Bloch phase, which satisfies the dispersion relation of the infinite crystal associated with $PCj$:

$$\cos \beta_j = \frac{1}{2} Tr(W_j) = f_j(\omega)\tag{3}$$

Now taking into account that

$$M = M_2 M_1 = W_2^{N_2} W_1^{N_1},\tag{4}$$

and combining this result with Equations (1) and (2), one finds the formula

$$\begin{aligned}\cos \beta \quad = \quad &K(\omega) = \frac{1}{2} Tr(W_2^{N_2} W_1^{N_1}) = K_1 \frac{1}{2} Tr(W_2 W_1)\\ &- K_2 \frac{1}{2} Tr(W_2) - K_3 \frac{1}{2} Tr(W_1) + K_4,\end{aligned}\tag{5}$$

for the dispersion relation of the combined structure $S$, where

$$\begin{aligned}K_1 \quad &= \quad \frac{\sin N_2 \beta_2 \sin N_1 \beta_1}{\sin \beta_2 \sin \beta_1},\\ K_2 \quad &= \quad \frac{\sin N_2 \beta_2 \sin(N_1 - 1)\beta_1}{\sin \beta_2 \sin \beta_1},\\ K_3 \quad &= \quad \frac{\sin(N_2 - 1)\beta_2 \sin N_1 \beta_1}{\sin \beta_2 \sin \beta_1},\\ K_4 \quad &= \quad \frac{\sin(N_2 - 1)\beta_2 \sin(N_1 - 1)\beta_1}{\sin \beta_2 \sin \beta_1}\end{aligned}\tag{6}$$

To obtain the explicit expression for $W_j$ with $j = 1, 2$, we focus on a binary PC whole unit cell consists of layers $A$ and $B$ with permittivity and permeability $(\varepsilon_a, \mu_a)$ and $(\varepsilon_b, \mu_b)$, respectively, as shown in Figure 1b, where the origin of the coordinates is chosen at the inversion center of layer $A$. The unit cell size or period is given by $d = a + b$, where $a$ and $b$ are the widths of layers $A$ and $B$, respectively. For this generic unit cell, it is straightforward to show that [22]

$$W = \begin{pmatrix} W_{11} & W_{12} \\ W_{21} & W_{22} \end{pmatrix} = \begin{pmatrix} 2ps - 1 & 2pq \\ 2rs & 2ps - 1 \end{pmatrix},\tag{7}$$

where

$$p = \cos\frac{1}{2}aQ_a \cos\frac{1}{2}bQ_b - \frac{\mu_b Q_a}{\mu_a Q_b}\sin\frac{1}{2}aQ_a \sin\frac{1}{2}bQ_b, \tag{8}$$

$$q = \frac{\mu_a}{Q_a}\sin\frac{1}{2}aQ_a \cos\frac{1}{2}bQ_b + \frac{\mu_b}{Q_b}\cos\frac{1}{2}aQ_a \sin\frac{1}{2}bQ_b, \tag{9}$$

$$r = -\frac{Q_b}{\mu_b}\cos\frac{1}{2}aQ_a \sin\frac{1}{2}bQ_b - \frac{Q_a}{\mu_a}\sin\frac{1}{2}aQ_a \cos\frac{1}{2}bQ_b, \tag{10}$$

$$s = \cos\frac{1}{2}aQ_a \cos\frac{1}{2}bQ_b - \frac{\mu_a Q_b}{\mu_b Q_a}\sin\frac{1}{2}aQ_a \sin\frac{1}{2}bQ_b. \tag{11}$$

In these equations,

$$Q_i = \sqrt{\frac{\omega^2}{c^2}n_i^2 - q^2}, \tag{12}$$

where $i = a, b$, $n_i = \sqrt{\varepsilon_i \mu_i}$ is the refractive index of layer $i$, and $q$ is the wave vector component of the electromagnetic wave. In the following, the analysis will be carried out for normal propagation, i.e., for $q = 0$.

Once the transfer matrix $M = W_2^{N_2} W_1^{N_1}$ through the combined structure $S$ is known, the derivation of the formula determining the complex transmission coefficient $t(\omega)$ can be found as follows. Limiting ourselves to the case where air is on both sides of $S$ and supposing that a monochromatic wave is incident from the left, as shown in Figure 1a, $t(\omega)$ satisfies the following relation [23]:

$$\begin{pmatrix} t(\omega) \\ 0 \end{pmatrix} = \widehat{T} \begin{pmatrix} 1 \\ r(\omega) \end{pmatrix}, \tag{13}$$

where $r(\omega)$ is the complex reflection coefficient and

$$\begin{aligned} \widehat{T} &= \begin{pmatrix} T_{11} & T_{12} \\ T_{21} & T_{22} \end{pmatrix} = S_0^{-1} W_2^{N_2} W_1^{N_1} S_0 \\ &= K_1 S_0^{-1}(W_2 W_1)S_0 - K_2 S_0^{-1} W_2 S_0 \\ &\quad - K_3 S_0^{-1} W_1 S_0 + K_4 I \end{aligned} \tag{14}$$

In this equation,

$$S_0 = \begin{pmatrix} 1 & 1 \\ iQ_0 & -iQ_0 \end{pmatrix} \tag{15}$$

with $Q_0 = \sqrt{\frac{\omega^2}{c^2}n_0^2 - q^2}$ being the positive component of the wavevector along the stacking direction in air ($n_0 = 1$). Using now Equation (13) and the fact that $\widehat{T}$ is a unimodular matrix, one immediately obtains the following expression for the transmission amplitude:

$$t(\omega) = \frac{1}{T_{22}}, \tag{16}$$

## 3. Results and Discussion

Let us use the above results to carry out a detailed study of the effects of the combined structure $S$ on the topological interface states. As we will see later, the Bloch phase $\beta$ in Equation (5) given by

$$\beta = \arccos[K(\omega)], \tag{17}$$

plays a fundamental role in the analysis of these effects. In the following, of course, we focus on $S = PC1 \mid PC2$ structures exhibiting topological interface modes, which, as mentioned above, are formed in the frequency range where the photonic bandgaps of the infinite crystals associated with $PC1$ and $PC2$ overlap. For one of these structures [8],

which will be used to illustrate our theoretical considerations, the $PC1$ parameters are $\varepsilon_{1b} = 3.8$, $\varepsilon_{1a} = \mu_{1a} = \mu_{1b} = 1$, $a_1 = 0.58d$, $b_1 = 0.42d$, whereas those of $PC2$ are $\varepsilon_{2b} = 4.2$, $\varepsilon_{2a} = \mu_{2a} = \mu_{2b} = 1$, $a_2 = 0.62d$, and $b_2 = 0.38d$, where $d$ is the period. Without loss of generality, the numerical calculations will be presented for $N_1 = N_2 = N$, where $N_1$ and $N_2$ are the number of unit cells of $PC1$ and $PC2$, respectively.

As shown in Ref. [8], this model structure exhibits a topological interface state localized in the spectral region 7 in Figure 2, where we have displayed the functions $f_1(\omega)$ and $f_2(\omega)$ in Equation (3), calculated from Equations (7)–(12). Notice that, ignoring some quantitative details, the curve $f_1(\omega)$ is essentially the same as that for $f_2(\omega)$. This means that since the frequency ranges for which $\left|f_j(\omega)\right| > 1$, with $j = 1, 2$, correspond to spectral regions of stop bands, the bandgaps of one of the infinite crystals overlap with those of another one, but only one of the overlapping regions support a localized mode. This can be verified by calculating the Zak phases associated with the band structure of the infinite version of $PC1$ and $PC2$ and noting that in region 7 the topological phase transition occurs, guaranteeing the existence of the interface mode. One can use two approaches to calculate the Zak phase $\theta_n$ of each isolated band $n$. In one of them, $\theta_n$ is expressed as an integral over the Brillouin zone in the reciprocal space of the Berry connection, whereas in the second one we can use the symmetry of the band edge states. In the latter, if the two band edge states of band $n$ have the same symmetry, $\theta_n = 0$, otherwise $\theta_n = \pi$. Using the symmetry approach and the fact that the zeros of the $\omega$-dependent functions in Equations (8)–(11) determine the symmetry properties of modes [24], it was found that $\theta_6 = \pi$ and $\theta_7 = 0$ for the infinite $PC1$ superlattice, while $\theta_6 = 0$ and $\theta_7 = \pi$ for another one, where $\theta_6$ and $\theta_7$ are, respectively, the Zak phases of the bands located below and above the gap in region 7 in Figure 2. Note the topological phase transition.

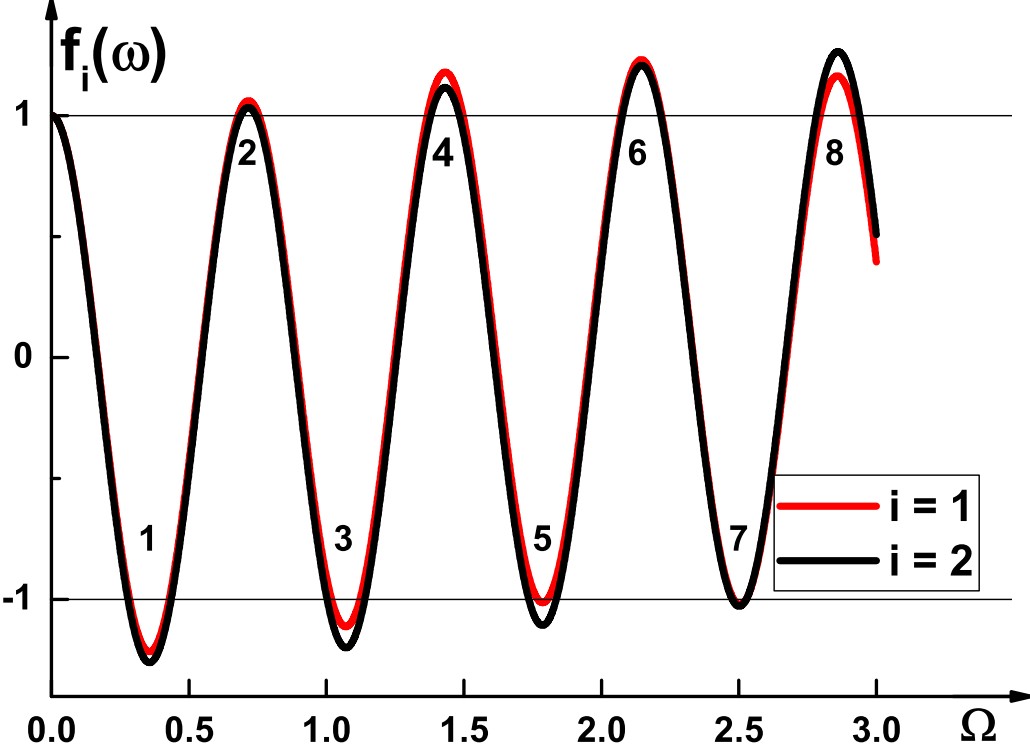

**Figure 2.** Functions $f_1(\omega) = \frac{1}{2}Tr(W_1)$ and $f_2(\omega) = \frac{1}{2}Tr(W_2)$ as functions of $\Omega = \omega d/2\pi c$, where $W_1$ and $W_2$ are the unit cell transfer matrices associated with the infinite version of $PC1$ and $PC2$, respectively. The $PC1$ parameters are $\varepsilon_{1b} = 3.8$, $\varepsilon_{1a} = \mu_{1a} = \mu_{1b} = 1$, $a_1 = 0.58d$, $b_1 = 0.42d$, whereas those of $PC2$ are $\varepsilon_{2b} = 4.2$, $\varepsilon_{2a} = \mu_{2a} = \mu_{2b} = 1$, $a_2 = 0.62d$, $b_2 = 0.38d$, where $d$ is the period. Note that the frequency regions where $|f_i(\omega)| > 1$, labeled by the numbers $1, 2, \ldots, 8$, determine the corresponding bandgaps.

Further, since the interface modes are formed in the combined structure $S = PC1 \mid PC2$, it is important to know how the bandgaps of the infinite crystals associated with $PC1$ and $PC2$ are reflected in the corresponding finite ones. To see this, we first use Equation (2) to obtain the expression

$$\cos K_j D_j = \frac{1}{2} Tr(W_j^{N_j}) = F_j(\omega) = \cos N_j \beta_j, \tag{18}$$

for the dispersion relation of $PCj$ with $j = 1, 2$, where $K_j$ and $D_j$ are the Bloch wavevector and width of photonic crystal $PCj$, and $\beta_j$ was introduced in Equation (3). It follows immediately from Equation (18) that the spectral regions where $\beta_j$ is a complex (real) quantity are quantitatively reproduced by $K_j$. That is, the spectral regions where the infinite version of $PCj$ exhibits bandgaps (pass bands) are exactly the same as those for $PCj$. This general result is illustrated in Figure 3 for the model structure considered above.

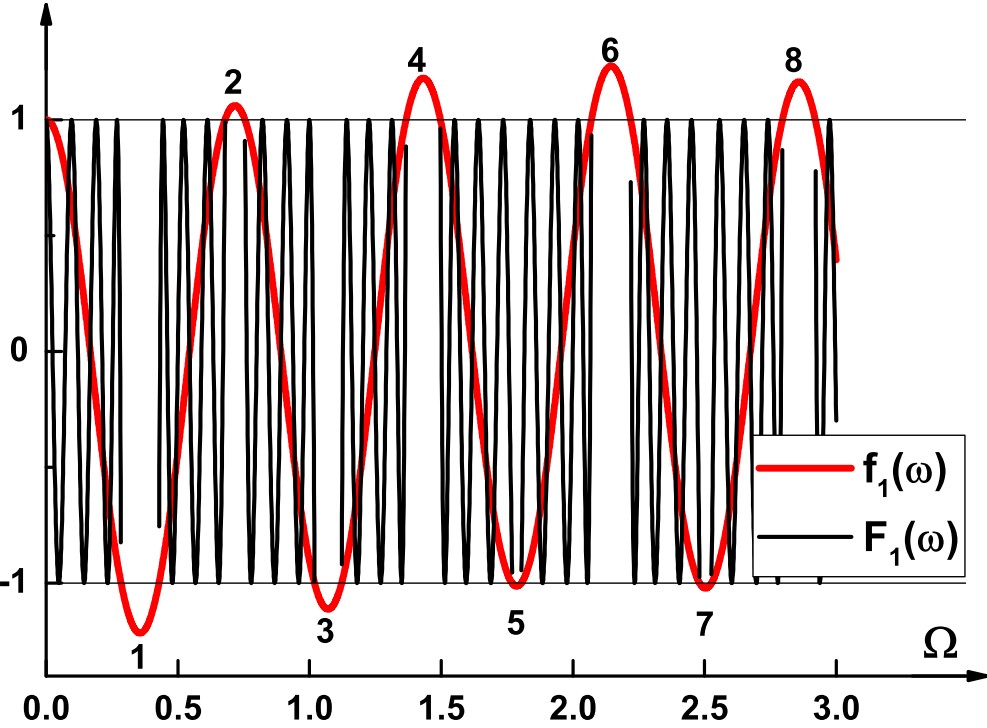

**Figure 3.** The same as in Figure 2, but substituting $f_2(\omega)$ with $F_1(\omega) = \frac{1}{2} Tr(W_1^{N_1})$ for $N_1 = 7$.

Furthermore, to compare the spectral distribution of bandgaps in the infinite crystals with that of the combined structure $S = PC1 \mid PC2$, the functions $f_1(\omega)$ in Equation (3) and $K(\omega)$ in Equation (5) are depicted in Figure 4a as functions of $\omega$. As one can see, both distributions are essentially the same, except in region 7 where $S$ exhibits the topological interface state. This suggests a certain correlation between the band structure of $S$ and the characteristics of these states, which will be investigated next. Of course, the analysis will be carried out in a certain frequency interval $\Delta$ including the frequency range where the mode is localized.

We begin by presenting in Figure 4b the results shown in Figure 4a, but for $\omega$ varying within the chosen interval $\Delta$. Note that in such an interval the band structure of $S$ exhibits two bands $B_1$ and $B_2$ sandwiched between two similar bandgaps labeled $\alpha$ and $\beta$. In order to understand the origin and properties of these bands, we have displayed in Figures 5–8 the dependence with $\omega$ (in units of $2\pi c/d$) of bands $B_1$ and $B_2$, the imaginary part of the Bloch phase $\beta(\omega)$, and the transmission spectrum $T(\omega)$, calculated from Equation (16) for increasing values of the number $N$ of unit cells. As one clearly sees in each one of these figures, (*i*) the bands $B_1$ and $B_2$ are separated by a minigap of width $\theta$ determining a

frequency range where $T(\omega)$ reaches its peak value and $Im[\beta(\omega)] > 0$. While these results can be understood as being due to topological effects, the inequality is a consequence of the fact that $\theta \neq 0$. (*ii*) The bands *B1* and *B2* are located in a frequency interval $\delta$ determined by the nearest edges of the similar $\alpha$ and $\beta$ bandgaps and are distributed in such a way that one is the mirror image of the other. This explains the $\omega$ location and properties of the transmission peak structure $T(\omega)$. In fact, when $\omega$ increases from left to right, $T(\omega)$ increases with $\omega$ within the *B1* passband and decreases inside the *B2* passband, forming the symmetric transmission peak structure observed in these figures.

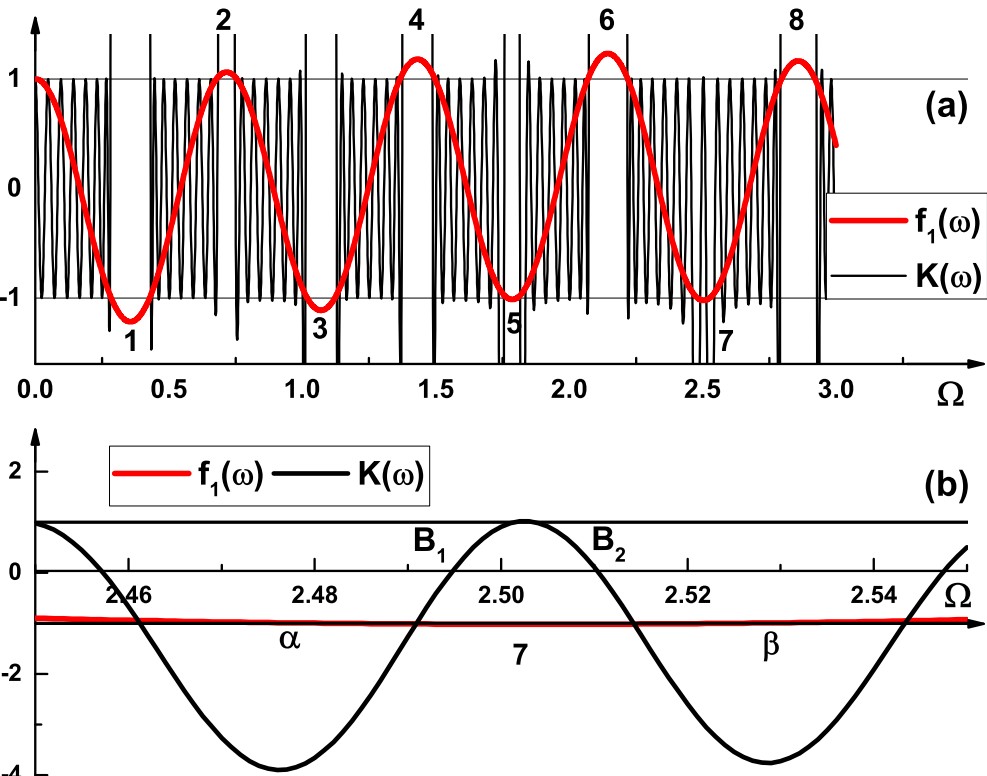

**Figure 4.** (**a**) The same as in Figure 2, but substituting $f_2(\omega)$ with $K(\omega) = \frac{1}{2}Tr(W_2^{N_2}W_1^{N_1})$ for $N_1 = N_2 = 7$. (**b**) The same as in (**a**), but for $\Omega$ varying within a smaller frequency interval $\Delta$ including the bandgap 7. Symbols $\alpha$ and $\beta$ label the corresponding bandgaps.

It is clear from Figures 5–8 that the interval $\delta$ is a decreasing function of the number $N$ of unit cells, leading to a reduction in the minigap width $\theta$ and, therefore, to a decrease in the transmission peak width. It is important to notice that these modifications are accompanied by a decrease in the resonant transmission peak, which tends to disappear for sufficiently large values of the number $N$ of unit cells. Physically, this behavior is a direct consequence of the fact that, for $N \gg 1$, the $\alpha$ and $\beta$ bandgaps come together, leading to the formation of a relatively large bandgap, which modifies the propagation of an electromagnetic wave through the combined structure substantially. Certainly, if the frequency of the incident wave is inside such a bandgap, it cannot propagate through the structure and will be completely reflected. Mathematically, since the resonant transmission peak is formed in the frequency range where the bandgaps of the infinite crystals associated with *PC1* and *PC2* overlap, the Bloch phases $\beta_1$ and $\beta_2$ in Equation (6) are complex quantities, which can be written as $i\phi$ and $\pi + i\phi$ at the Brillouin zone center and edge, respectively, where $\phi$ is a real angle. Using these expression for $\beta_1$ and $\beta_2$ in Equations (6) and (14)–(16), it is straightforward to show that the transmission spectrum $T(\omega) = |t(\omega)|^2$ decreases exponentially with the number $N$ of unit cells for $N|\phi| \gg 1$.

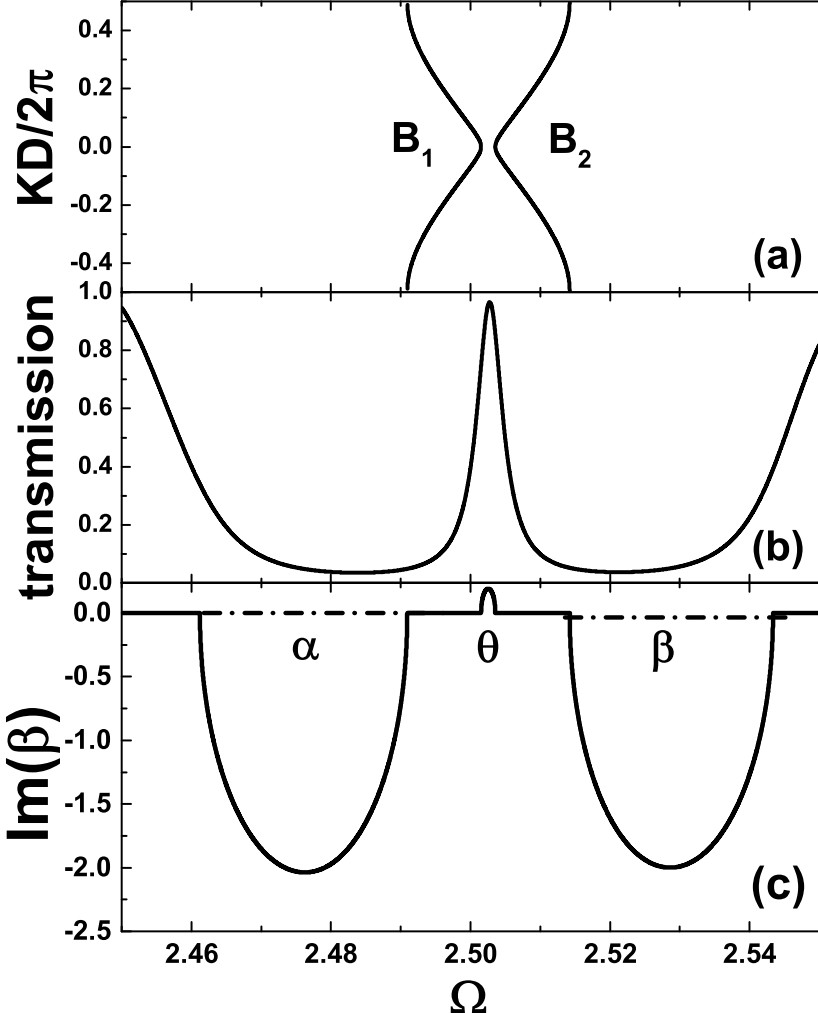

**Figure 5.** (**a**) Bands, (**b**) frequency-dependent transmission spectrum, and (**c**) imaginary part of Bloch phase of the combined structure $S$, for $N_1 = N_2 = 7$ and $\Omega = \omega d / 2\pi c$ varying within the frequency interval $\Delta$ including the bandgap 7. Symbols $\alpha$ and $\beta$ as in Figure 4b.

We have added two insets in panel (b) of Figures 7 and 8, where we present a zoom-in of the central peak close to $\Omega = 2.5$. The zoom-in has been performed to the point of determining that there is no longer a way to hide the height of the transmission peak. Note that as the value of $N$ increases, the width of the central peak near $\Omega = 2.5$ becomes smaller, and its height decreases, as expected.

It should be added, finally, that while the topological phase transition occurring between *PC*1 and *PC*2 guarantees the existence of the topological interface modes [8], the configuration of bands and bandgaps discussed above determines the physical mechanism supporting them and the properties of the corresponding resonant transmission peaks, whose characteristics can be controlled by the number $N$ of unit cells. Such a control can be used to design structures with the desired features.

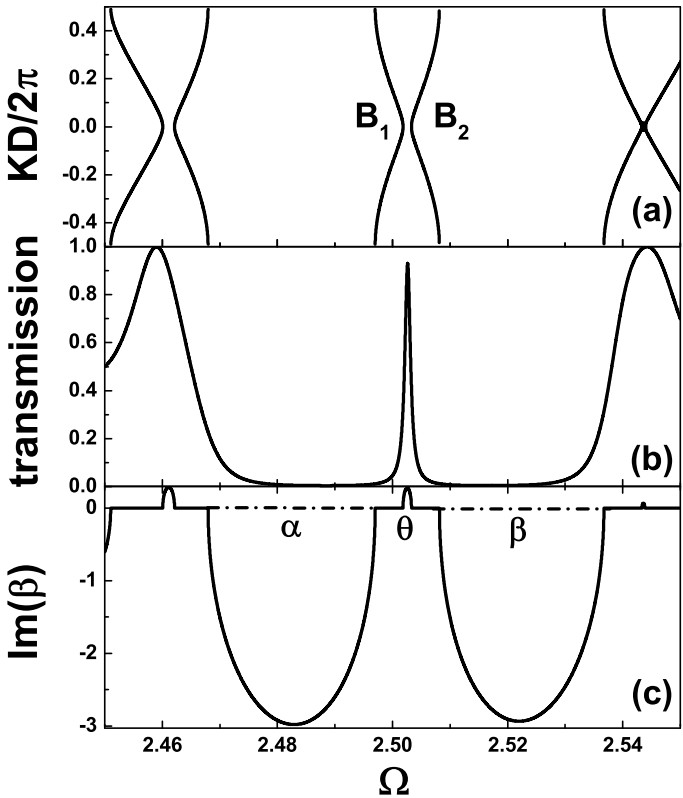

**Figure 6.** The same as in Figure 5, but for $N_1 = N_2 = 10$.

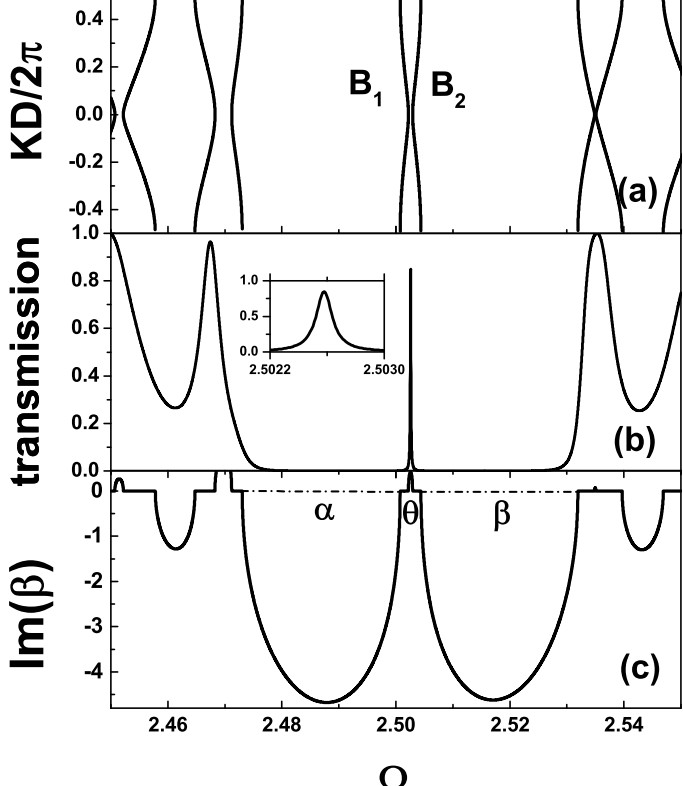

**Figure 7.** The same as in Figure 5, but for $N_1 = N_2 = 15$. The inset in (**b**) shows a zoom-in of the transmission peak located at $\Omega = 2.5026$.

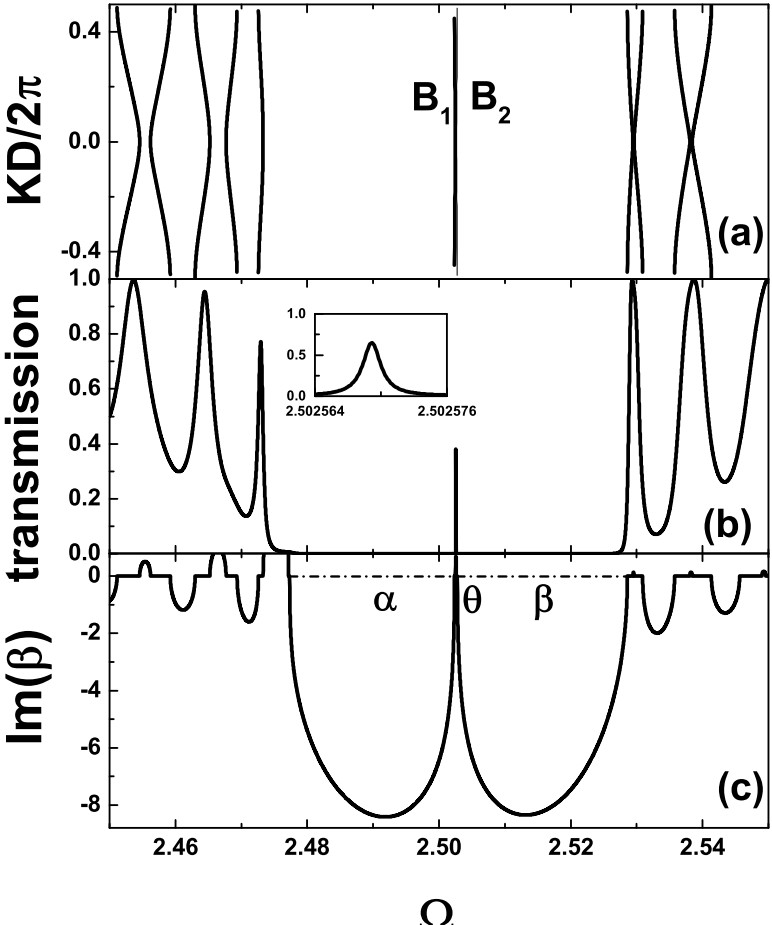

**Figure 8.** The same as in Figure 5, but for $N_1 = N_2 = 25$. The inset in (**b**) shows a zoom-in of the transmission peak located at $\Omega = 2.502570$.

## 4. Conclusions

We have presented a detailed study of the physical mechanisms determining the existence of topological interface modes in photonic structures. It was shown that these mechanisms are directly correlated with the formation of bands and bandgaps of topological origin in these structures, which can be used to explain the main characteristics of the resonant transmission peaks associated with such modes. It was demonstrated that the width and maximum value of these transmission peaks decrease when the number $N$ of unit cells increases. Consequently, such peaks tend to disappear for sufficiently large values of $N$.

Finally, it is not difficult to verify that the results obtained in this work are general enough and may be used to describe and understand the properties of the topological interface modes in a wide variety of photonic structure.

**Author Contributions:** M.d.D.-L.: conceptualization, methodology, software, formal analysis, investigation, supervision, and writing; A.M.-G.: methodology, software, and formal analysis; C.A.D.: formal analysis and writing. All authors have read and agreed to the published version of the manuscript.

**Funding:** C.A.D. is grateful to the Colombian agencies CODI-Universidad de Antioquia (Estrategia de Sostenibilidad de la Universidad de Antioquia and projects "Propiedades magneto-ópticas y óptica no lineal en superredes de Grafeno", "Estudio de propiedades ópticas en sistemas semiconductores de dimensiones nanoscópicas", "Propiedades de transporte, espintrónicas y térmicas en el sistema molecular ZincPorfirina", and "Complejos excitónicos y propiedades de transporte en sistemas nanométricos de semiconductores con simetría axial"), and Facultad de Ciencias Exactas y Naturales-Universidad de Antioquia (CAD exclusive dedication project 2022–2023).

**Institutional Review Board Statement:** Not applicable.

**Informed Consent Statement:** Not applicable.

**Data Availability Statement:** No new data were created or analyzed in this study. Data sharing is not applicable to this article.

**Conflicts of Interest:** The authors declare no conflicts of interest.

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
