# Peer review of "Exploring Photonic Crystals: Band Structure and Topological Interface States"

_condensedmatter, doi:10.3390/condmat8030063_

Round 1
Reviewer 1 Report
In this manuscript, the authors discuss the existence and determining the characteristics of topological interface states in one-dimensional photonic structure. It is already known that the physical mechanisms supporting topological interface states are directly related to the different topological property of bandgaps in these structures. We know that the periodicity causes the bandgap, and the calculated result for 1/2Tr(W2N2 W1N1) in the manuscript is that for a structure with a finite-period number, which is similar to the transmission of the finite-size structure. Thus, using the term “B1 and B2 subbands” in the manuscript is inappropriate. Another problem is that the width and maximum value of the transmission peaks decrease when the number N of unit cells increases. In general, when the number N of unit cells increases, the line width of the transmission spectra decreases, but the value of the transmission peak remains unchanged as long as N1=N2 (zero-reflection condition). This result might be caused by a broad frequency interval used in the transmission spectra calculations. The frequency interval should be very small when the transmission is very sharp in order to capture the transmission peak.
Overall, the results are misleading in the manuscript at present, and I would not recommend its publication in present version.
Author Response
Comments of Referee 1
The Referee:
In this manuscript, the authors discuss the existence and determining the characteristics of topological interface states in one-dimensional photonic structure. It is already known that the physical mechanisms supporting topological interface states are directly related to the different topological property of bandgaps in these structures. We know that the periodicity causes the bandgap, and the calculated result for 1/2Tr(W2N2W1N1) in the manuscript is that for a structure with a finite-period number, which is similar to the transmission of the finite-size structure. Thus, using the term “B1 and B2 subbands” in the manuscript is inappropriate.
Our reply:
The paragraph beginning with the word “Since” in page 2 was replaced by a new paragraph which reads:
“To obtain the energy spectrum of the finite combined structure S, we adopt the procedure in which S is taken as the unit cell of a 1D infinite periodic superlattice. As shown in different calculations \cite{20,21}, this procedure describes appropriately the energy spectrum of any finite structure, especially when its size is large enough”.
The Referee:
Another problem is that the width and maximum value of the transmission peaks decrease when the number N of unit cells increases. In general, when the number N of unit cells increases, the line width of the transmission spectra decreases, but the value of the transmission peak remains unchanged as long as N1=N2 (zero-reflection condition). This result might be caused by a broad frequency interval used in the transmission spectra calculations. The frequency interval should be very small when the transmission is very sharp in order to capture the transmission peak.
Our reply:
At the end of the paragraph following Fig. 5, we have added the following comment:
“Mathematically, since the resonant transmission peak is formed in the frequency range where the bandgaps of the infinite crystals associated with $PC1$ and $PC2$ overlap, the Bloch phases $\beta_{1}$ and $\beta_{2}$ in Eq. (6) are complex quantities, which can be written as $i\,\phi$ and $\pi+i\,\phi$ at the Brillouin zone center and edge, respectively, where $\phi$ is a real angle. Using these expression for $\beta_{1}$ and $\beta_{2}$ in Eqs. (6) and (14)-(16), it is straightforward to show that the transmission spectrum $T(\omega)=\left\vert t(\omega)\right\vert^{2}$ decreases exponentially with the number $N$ of unit cells for $N\left\vert \phi \right\vert >>1$.”
The Referee:
Overall, the results are misleading in the manuscript at present, and I would not recommend its publication in present version.
Our reply:
We hope that the Referee finds our answers satisfactory and that he/she considers that the revised version of our manuscript is suitable for publication in the Condensed Matter Journal.

Reviewer 2 Report
In this work, the authors proposed a theoretical study of the physical mechanisms determining the existence of topological interface modes in photonic structures, formed with the concatenation of two finite, N -period, one-dimensional photonic crystals. Meanwhile, the formation of subbands and bandgaps of topological origin in these structures is investigated. In general, the idea of the manuscript appears to be interesting, and the numerical findings may be promising. However, I believe that the current version needs for MAJOR revisions to be suitable for publication. In this regard, the authors should carefully address the following points: -
1- In general, the language level in writing this manuscript needs for some revisions.
2- The abstract must be rewritten and reorganized.
3- I believe that the introduction section is not satisfactory. Notably, some of the fundamental concepts are missing. The authors should describe in detail the basic concepts of PCs and their role in some different applications.
4- The novelty of the presented study over its counterparts is missing. You should highlight the novelty of your work over others. In particular, the idea of topological edge states is extensively discussed.
5- The authors should indicate the advantages of considering such design over the defective one from manufacturing and cost levels.
6- The physical explanations behind the emergence of the resonant mode inside the photonic bandgap are missing.
In general, the language level in writing this manuscript needs for some revisions.
Author Response
Comments of Referee 2
In this work, the authors proposed a theoretical study of the physical mechanisms determining the existence of topological interface modes in photonic structures, formed with the concatenation of two finite, N -period, one-dimensional photonic crystals. Meanwhile, the formation of subbands and bandgaps of topological origin in these structures is investigated. In general, the idea of the manuscript appears to be interesting, and the numerical findings may be promising. However, I believe that the current version needs for MAJOR revisions to be suitable for publication. In this regard, the authors should carefully address the following points: -
The Referee:
1-In general, the language level in writing this manuscript needs for some revisions.
Our reply:
We have carried out a detailed English revision
The Referee:
2-The abstract must be rewritten and reorganized.
Our reply:
The abstract was rewritten
The Referee:
3-I believe that the introduction section is not satisfactory. Notably, some of the fundamental concepts are missing. The authors should describe in detail the basic concepts of PCs and their role in some different applications.
Our reply:
The first part of the INTRODUCTION, beginning with the word “Since”, was replaced by another one, which reads:
“Since the seminal work of Haldane and Raghu \cite{1}, many theoretical and experimental works have been devoted to the study of topological phenomena in photonic crystals \cite{2}, which are artificial periodic arrays of materials with different refractive indices in one, two, and three dimensions. This interest is motivated by the fact that photonic crystals offer the possibility of controlling and manipulating\ the properties of light, a possibility closely related to the existence of photonic bandgaps in the dispersion relation of these structures. Thus, if the frequency of a light beam is in the range of a bandgap, the beam cannot propagate inside the structure, leading to remarkable features that have applications in many photonic devices, such as optical isolators, topological lasers, tunable filters, and resonators \cite{3,4,5,6}.”
The Referee:
4-The novelty of the presented study over its counterparts is missing. You should highlight the novelty of your work over others. In particular, the idea of topological edge states is extensively discussed.
Our reply:
The last paragraph of the INTRODUCTION was replaced by a new paragraph, which reads:
“In contrast to the existent approach, in which the condition for the existence of an interface state has been established by using the topological properties of the constituting PCs, here we focus on the role played by the band structure of the concatenated PCs. It is shown then that the physical mechanism supporting the existence of the interface states originates from a specific configuration of subbands and bandgaps of topological origin in the mentioned band structure. This result not only provides a novel understanding of the topological interface states but also provides an efficient procedure for controlling their properties, as will be shown below.”
The Referee:
5-The authors should indicate the advantages of considering such design over the defective one from manufacturing and cost levels.
Our reply:
In the paragraph located just before the “Conclusions” an additional text was included. It reads:
“It should be added, finally, that while the topological phase transition occurring between $PC1$ and $PC2$ guarantees the existence of the topological interface modes \cite{6}, the configuration of subbands and bandgaps discussed above determines the physical mechanism supporting them and the properties of the corresponding resonant transmission peak, whose characteristics can be controlled by the number $N$ of unit cells. Such a control can be used to design structures with the desired features.”
The Referee:
6-The physical explanations behind the emergence of the resonant mode inside the photonic bandgap are missing.
Our reply:
At the end of the paragraph located before Fig. 5, we added the following text which modifies the paragraph in the former version of the manuscript:
“The subbands $B1$ and $B2$ are located in a frequency interval $\delta$ determined by the nearest edges of the similar $\alpha $ and $\beta $ bandgaps and are distributed in such a way that one is the mirror image of the other. This explains the $\omega$-location and properties of the transmission peak structure $T(\omega)$. In fact, when $\omega$ increases from left to right, $T(\omega)$ increases with $\omega$ within the $B1$-passband and decreases inside the $B2$-passband, forming the symmetric transmission peak structure observed in these figures.”
We hope that the Referee finds our answers satisfactory and that he/she considers that the revised version of our manuscript is suitable for publication in the Condensed Matter Journal.

Reviewer 3 Report
In this paper, band structure and topological interface states of one-dimensional photonic crystal with two-different finite period photonic crystal.
The analysis method is based on transfer matrix method which is usually used for the analysis of one-dimensional photonic crystal. That is, analysis method is not novel. The transmission property of one-dimensional photonic crystal is investigated for different number of period. Transmission band become narrower when number of periods is increased. This is an usual behavior. However, when N1 = N2 > 25, transmission band tends to be disappeared. I am wondering if sampling points along horizontal axis is really sufficient. That is, I wonder if the peak is just skipped. Please discuss why this phenomena of disappearing transmission band appear in more detail.
Author Response
Comments of Referee 3
In this paper, band structure and topological interface states of one-dimensional photonic crystal with two-different finite period photonic crystal.
The Referee:
The analysis method is based on transfer matrix method which is usually used for the analysis of one-dimensional photonic crystal. That is, analysis method is not novel. The transmission property of one-dimensional photonic crystal is investigated for different number of period. Transmission band become narrower when number of periods is increased. This is a usual behavior. However, when N1 = N2 > 25, transmission band tends to be disappeared. I am wondering if sampling points along horizontal axis is really sufficient. That is, I wonder if the peak is just skipped. Please discuss why this phenomena of disappearing transmission band appear in more detail.
Our Reply:
At the end of the paragraph after Fig. 3 we have added the following comment:
“Mathematically, since the resonant transmission peak is formed in the frequency range where the bandgaps of the infinite crystals associated with $PC1$ and $PC2$ overlap, the Bloch phases $\beta_{1}$ and $\beta_{2}$ in Eq. (6) are complex quantities, which can be written as $i\,\phi$ and $\pi+i\,\phi$ at the Brillouin zone center and edge, respectively, where $\phi$ is a real angle. Using these expression for $\beta_{1}$ and $\beta_{2}$ in Eqs. (6) and (14)-(16), it is straightforward to show that the transmission spectrum $T(\omega)=\left\vert t(\omega)\right\vert^{2}$ decreases exponentially with the number $N$ of unit cells for $N\left\vert \phi \right\vert >>1$.”
We hope that the Referee finds our answers satisfactory and that he/she considers that the revised version of our manuscript is suitable for publication in the Condensed Matter Journal.

Reviewer 4 Report
The paper is devoted to investigation of optical properties of 1D photonic crystals (PC) in the case of the concatenation of two PC. The authors presented zone structure of the whole system for different important practical cases.
The theoretical description, text, figures, literature correspnd to the journal profile and requirement in general. It is to add some references on similar cases of multilayer structures with different materials (dielectrics, metals, semiconductors), e.g., a very recent paper "Belyaev, V.; Zverev, N.; Abduev, A.; Zotov, A. E-Wave Interaction with the OneDimensional Photonic Crystal with Weak Conductive and Transparent Materials. Coatings 2023, 13, 712. https://doi.org/10.3390/ coatings13040712".
Author Response
Comments of Referee 4
The Referee:
The paper is devoted to investigation of optical properties of 1D photonic crystals (PC) in the case of the concatenation of two PC. The authors presented zone structure of the whole system for different important practical cases.
The theoretical description, text, figures, literature correspond to the journal profile and requirement in general. It is to add some references on similar cases of multilayer structures with different materials (dielectrics, metals, semiconductors), e.g., a very recent paper "Belyaev, V.; Zverev, N.; Abduev, A.; Zotov, A. E-Wave Interaction with the OneDimensional Photonic Crystal with Weak Conductive and Transparent Materials. Coatings 2023, 13, 712. https://doi.org/10.3390/ coatings13040712".
Our Replay
1) The suggested Reference has been included, see Ref. [19] in the revised version of the manuscript.
We hope that the Referee finds our answers satisfactory and that he/she considers that the revised version of our manuscript is suitable for publication in the Condensed Matter Journal.

Round 2
Reviewer 1 Report
In the revised manuscript, the author has improved the work in a few ways. However, there are still many problems in the manuscript.
It is know that the topological interface states originated from the different topological characteristics of the bandgaps in photonic crystals rather than the subbands. Thus, it is misleading to calculate the structure's subbands using the heterostructure S as a super unit cell. Another problem is that the property of the transmission spectrum can not be intuitively seen according to Eqs. 6 and (14)-(16). Hence, there is no reasonable explanation for the decrease in the peak value of the transmission spectrum. It can see that from Fig. 1(b), the unit cell of the structure is symmetrical. Usually, when the number N of unit cells increases, the line width of the transmission spectra decreases, but the value of the transmission peak remains unchanged as long as N1=N2 (zero-reflection condition). Therefore, the calculated results are confusing.
Overall, I would not recommend its publication in the Condensed Matter Journal.
Reviewer 2 Report
The present form of this paper can be accepted for publication.
Author Response
Thanks for your comments.
Round 3
Reviewer 1 Report
The revised manuscript has made significant improvements at present. However, there are still some minor problems in the manuscript.
The calculated Zak phases are directly related to the reflection phases of the band gaps, as shown in Eq. D12 in Ref. 8. Namely, the topological interface states originated from the different topological characteristics of the band gaps in photonic crystals rather than the bands. Besides, we also accept that the transmittance of the topological interface states may not reach 1. Another problem is that the manuscript still has some formatting issues that need to be carefully checked. For instance, Figs. 5-8's figure format should be uniform, and several references' abbreviations are erroneous.
Overall, I can recommend publishing on the Journal Condensed Matter after addressing the problems above.
